# HEDNet: A Hierarchical Encoder-Decoder Network for 3D Object Detection in Point Clouds

**Gang Zhang**[1], **Junnan Chen**[2], **Guohuan Gao**[3], **Jianmin Li**[1], **Xiaolin Hu**[1,4,5*]

[1]Department of Computer Science and Technology, Institute for AI,
BNRist, THU-Bosch JCML Center, Tsinghua University
[2]Huazhong University of Science and Technology, [3]Beijing Institute of Technology
[4]Tsinghua Laboratory of Brain and Intelligence (THBI),
IDG/McGovern Institute for Brain Research, Tsinghua University
[5] Chinese Institute for Brain Research (CIBR), Beijing 100010, China
zhang-g19@mails.tsinghua.edu.cn,chen_jn@hust.edu.cn,gaoguohuan@bit.edu.cn
lijianmin@mail.tsinghua.edu.cn,xlhu@tsinghua.edu.cn
Code: https://github.com/zhanggang001/HEDNet

## Abstract

3D object detection in point clouds is important for autonomous driving systems. A primary challenge in 3D object detection stems from the sparse distribution of points within the 3D scene. Existing high-performance methods typically employ 3D sparse convolutional neural networks with small kernels to extract features. To reduce computational costs, these methods resort to submanifold sparse convolutions, which prevent the information exchange among spatially disconnected features. Some recent approaches have attempted to address this problem by introducing large-kernel convolutions or self-attention mechanisms, but they either achieve limited accuracy improvements or incur excessive computational costs. We propose HEDNet, a hierarchical encoder-decoder network for 3D object detection, which leverages encoder-decoder blocks to capture long-range dependencies among features in the spatial space, particularly for large and distant objects. We conducted extensive experiments on the Waymo Open and nuScenes datasets. HEDNet achieved superior detection accuracy on both datasets than previous state-of-the-art methods with competitive efficiency. The code has been released.

## 1 Introduction

Learning effective representations from sparse input data is a key challenge for 3D object detection in point clouds. Existing point-based methods [1, 2, 3, 4, 5] and range-based methods [6, 7, 8, 9, 10] either suffer from high computational costs or exhibit inferior detection accuracy. Currently, voxel-based methods [11, 12, 13, 14, 15] dominate high-performance 3D object detection.

The voxel-based methods partition the unstructured point clouds into regular voxels and utilize sparse conventional neural network (CNNs) [11, 12, 16, 17, 18, 19] or transformers [13, 14, 15] as backbones for feature extraction. Most existing sparse CNNs are primarily built by stacking submanifold sparse residual (SSR) blocks, each consisting of two submanifold sparse convolutions [20] with small kernels. However, submanifold sparse convolutions maintain the same sparsity between input and output features, and therefore hinder the exchange of information among spatially disconnected features. Consequently, models employing SSR blocks face challenges in effectively capturing long-range dependencies among features. One potential solution is to replace the submanifold

---

*Corresponding Author

37th Conference on Neural Information Processing Systems (NeurIPS 2023).

sparse convolutions in SSR block with regular sparse convolutions [21]. However, this leads to a significant decrease in feature sparsity as the network deepens, resulting in substantial computational costs. Recent research has investigated the utilization of large-kernel sparse CNNs [12, 16] and transformers [14, 15] to capture long-range dependencies among features. However, these approaches have either demonstrated limited improvements in detection accuracy or come with significant computational costs. Thus, the question remains: *is there an efficient method that enables sparse CNNs to effectively capture long-range dependencies among features?*

Revisiting backbone designs in various dense prediction tasks [13, 22, 23, 24, 25, 26], we observe that the encoder-decoder structure has proven effective in capturing long-range dependencies among features. These methods typically use a high-to-low resolution backbone as an encoder to extract multi-scale features and design different decoders to recover high-resolution features that can model long-range relationships. For instance, PSPNet [24] incorporates a pyramid pooling module to capture both local and global contextual information by pooling features at multiple scales. SWFormer [13] integrates a top-down pathway into its transformer backbone to capture cross-window correlations. However, the utilization of the encoder-decoder structure in designing sparse convolutional backbones for 3D object detection has not yet been explored, to the best of our knowledge.

In this work, we propose a sparse encoder-decoder (SED) block to overcome the limitations of the SSR block. The encoder extracts multi-scale features through feature down-sampling, facilitating information exchange among spatially disconnected regions. Meanwhile, the decoder incorporates multi-scale feature fusion to recover the lost details. A hallmark of the SED block is its ability to capture long-range dependencies while preserving the same sparsity between input and output features. Since current leading 3D detectors typically rely on object centers for detection [27, 28], we further adapt the 3D SED block into a 2D dense encoder-decoder (DED) block, which expands the extracted sparse features towards object centers. Leveraging the SED block and DED block, we introduce a hierarchical encoder-decoder network named HEDNet for 3D object detection in point clouds. HEDNet can learn powerful representations for the detection of large and distant objects.

Extensive experiments were conducted on the challenging Waymo Open [29] and nuScenes [30] datasets to demonstrate the effectiveness of the proposed HEDNet on 3D object detection. HEDNet achieved impressive performance, with a 75.0% L2 mAPH on the Waymo Open *test* set and a 72.0% NDS on the nuScenes *test* set, outperforming prior methods that utilize large-kernel CNNs or transformers as backbones while exhibiting higher efficiency. For instance, HEDNet was 50% faster than DSVT, the previous state-of-the-art transformer-based method, with 1.3% L2 mAPH gains.

## 2 Related work

### 2.1 3D object detection in point clouds

For 3D object detection in point clouds, methods can be categorized into three groups: point-based, range-based, and voxel-based. Point-based methods [1, 2, 3, 4, 5] utilize the PointNet series [31, 32] to directly extract geometric features from raw point clouds and make predictions. However, these methods require computationally intensive point sampling and neighbor search procedures. Range-based methods [6, 7, 8, 9, 10] convert point clouds into pseudo images, thus benefiting from the well-established designs of 2D object detectors. While computationally efficient, these methods often exhibit lower accuracy. Voxel-based approaches [11, 17, 18, 19] are currently the leading methods for high-performance 3D object detection. Most voxel-based methods employ sparse CNNs that consist of submanifold and regular sparse convolutions with small kernels to extract features. Regular sparse convolutions can capture distant contextual information but are computationally expensive. On the other hand, submanifold sparse convolutions prioritize efficiency but sacrifice the model's ability to capture long-range dependencies.

### 2.2 Capturing long-range dependencies for 3D object detection

To capture long-range dependencies for 3D object detection, recent research has explored solutions such as large-kernel sparse convolutions [33, 16] and self-attention mechanisms [13, 14, 15]. However, directly applying plain large-kernel CNNs for 3D representation learning can lead to problems such as overfitting and reduced efficiency. Weight-sharing strategies have been proposed to mitigate overfitting, like LargeKernel3D [12] and Link [16], however, they still suffer from low efficiency.

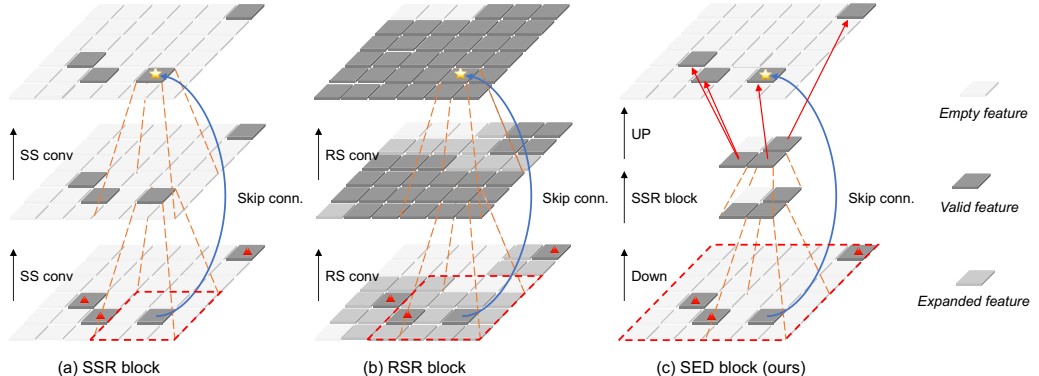

Figure 1: Comparison among SSR block (a), RSR block (b), and our SED block (c). The 'Skip conn.' denotes the skip connection, and the orange dashed lines represent the convolution kernel space. Valid features have non-zero values. Expanded and empty features have zero values. In (b), convolution is applied to both valid and expanded features, *i.e.,*the convolution kernel center traverses the regions covered by these features. The red dashed square highlights the regions from which the output feature marked by a star can receive information. In (c), we adopt a 3×3 RS convolution *with a stride of 3* for feature down-sampling (Down) as an example. UP denotes feature up-sampling.

Other methods, such as SST [14] and DSVT [15], utilize transformers as replacements for sparse CNNs. SST employs a single-stride sparse transformer to preserve high-resolution features without using down-sampling operators that may cause a loss of detail. Similarly, DSVT employs a single-stride sparse transformer and performs window-based self-attention sequentially along the X-axis and Y-axis. While both large-kernel CNNs and transformers aim to capture long-range dependencies, they either achieve comparable performance or exhibit lower efficiency compared with sparse CNNs. In contrast, our proposed HEDNet effectively captures long-range dependencies with the help of encoder-decoder blocks while achieving competitive inference speed compared with existing methods.

## 2.3 Encoder-decoder networks for dense prediction

The encoder-decoder structure has been extensively investigated in various dense prediction tasks. For example, the FPN-series [22, 34, 35, 36] incorporates lightweight fusion modules as decoders to integrate multi-scale features from image classification backbones. DeeplabV3+ [25] employs an atrous spatial pyramid pooling module to combine low-level features with semantically rich high-level features. SWFormer [13] introduces a top-down pathway into its transformer backbone to capture cross-window correlations. However, there is limited exploration of the encoder-decoder structure in the design of sparse CNNs. Most voxel-based approaches [17, 18, 27, 28] rely on high-to-low resolution sparse CNNs to extract single-scale high-level features. Part-A2-Net [37] adopts the UNet [38] to extract features, but it performs detection using the encoder of UNet while utilizing the decoder for the auxiliary segmentation and part prediction tasks. In this study, we propose HEDNet, which primarily consists of encoder-decoder blocks to effectively capture long-range dependencies.

## 3 Method

### 3.1 Background

The sparse CNNs adopted by most voxel-based methods [18, 27, 28] are primarily built by stacking SSR blocks, each consisting of two submanifold sparse (SS) convolutions [20]. In addition, they usually insert regular sparse (RS) convolutions [21] into the stacked SSR blocks to reduce the resolution of feature maps progressively (similar to ResNet [39]).

**SS convolution and SSR block.** We present the structure of a single SSR block in Figure 1 (a). Two SS convolutions are sequentially applied to the input feature map, with skip connections incorporated between the input and output feature maps of the SSR block. The *sparsity of feature map* is defined as the ratio of the regions that are **not** occupied by valid (nonzero) features to the total area of the feature map. SS convolution only operates on valid features, allowing the output feature map of the SSR block to maintain the same sparsity as the input feature map. However, this design hinders the

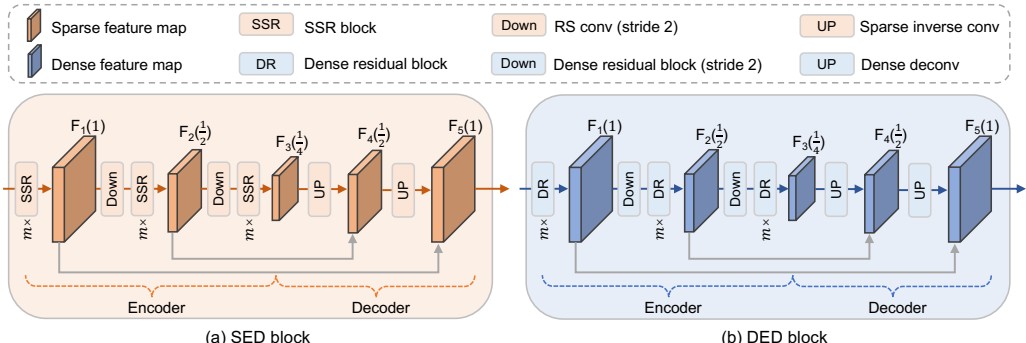

Figure 2: Architecture of the SED block (a) and DED block (b). As an example, we illustrate blocks of three scales. Both designs share the same structure. $F_1$/$F_2$/$F_3$/$F_4$/$F_5$ are the names of the corresponding feature maps. The number in parentheses indicates the resolution ratio of the corresponding feature map relative to the block input. The SED block is capable of processing both 2D and 3D features, depending on whether 2D or 3D sparse convolutions are used.

exchange of information among spatially disconnected features. For instance, in the top feature map, the output feature marked by a star cannot receive information from the other three feature points outside the red dashed square in the bottom feature map (marked by the red triangles). This poses a challenge for the model in capturing long-range dependencies.

**RS convolution and RSR block.** One possible solution to problem is to replace the SS convolutions in the SSR block with RS convolutions. We call this modified structure regular sparse residual (RSR) block and illustrate its structure in Figure 1 (b). RS convolution operates on both valid and expanded features [21]. Expanded features correspond to the features that fall within the neighborhood of the valid features. Taking a 2D RS convolution with a kernel size of $3 \times 3$ as an example, the neighborhood of a certain valid feature consists of the eight positions around it. This design leads to an output feature map with a lower sparsity compared with the input feature map. Stacking RS convolutions reduces the feature sparsity dramatically, which in turn leads to a notable decrease in model efficiency compared with using SS convolutions. This is why existing methods [18, 27, 28] typically limit the usage of RS convolution to feature down-sampling layers.

## 3.2 SED and DED blocks

**SED block.** SED block is designed to overcome the limitations of SSR block. The fundamental idea behind this design is to reduce the spatial distance between distant features through feature down-sampling and recover the lost details through multi-scale feature fusion.

We illustrate a two-scale SED block in Figure 1 (c). After feature down-sampling, the spatially disconnected valid features in the bottom feature map are integrated into the adjacent valid features in the middle feature map. An SSR block is subsequently applied to the middle feature map to promote interaction among valid features. Finally, the middle feature map is up-sampled to match the resolution of the input feature map. *Note that the feature up-sampling layer (UP) only up-samples features to the regions covered by the valid features in the input feature map.* As a result, the proposed SED block can maintain the same sparsity between input and output feature maps. This characteristic prevents the introduction of excessive computational costs when stacking multiple SED blocks.

The architecture of a three-scale SED block is presented in Figure 2 (a). The SED block adopts an asymmetric encoder-decoder structure similar to UNet [38], with the encoder responsible for extracting multi-scale features and the decoder sequentially fusing the extracted multi-scale features with the help of skip connections. Given the input feature map X, the function of the SED block can be formulated as follows:

$$F_1 = SSR^m(X) \tag{1}$$

$$F_2 = SSR^m(Down_1(F_1)) \tag{2}$$

$$F_3 = SSR^m(Down_2(F_2)) \tag{3}$$

$$F_4 = UP_2(F_3) + F_2 \tag{4}$$

$$F_5 = UP_1(F_4) + F_1 \tag{5}$$

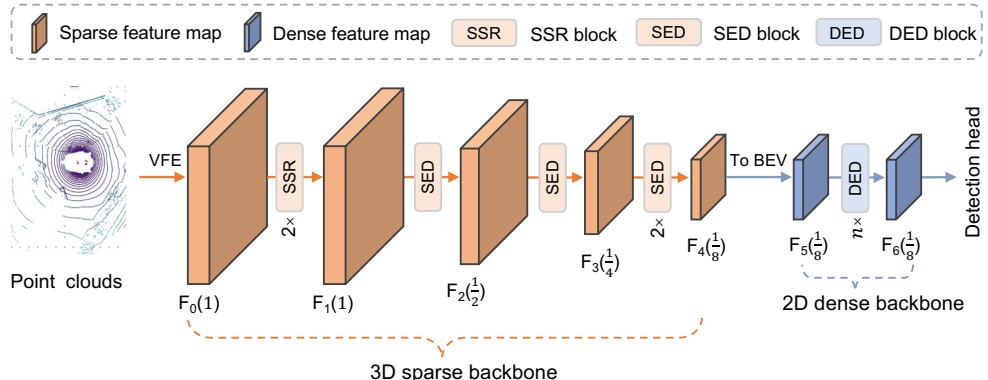

Figure 3: Architecture of the proposed HEDNet. Given the raw point clouds, we first perform voxelization to generate voxels by the VFE module, then employ the 3D sparse backbone and the 2D dense backbone to extract features for the detection head. The number in the bracket denotes the resolution ratio of the corresponding feature map relative to the input. *The RS convolutions for feature down-sampling that follow the feature maps $F_1$, $F_2$, and $F_3$ are omitted for simplicity.*

where $F_5$ denotes the output feature map with the same resolution as the input X. The resolution ratios of the intermediate feature maps $F_1$, $F_2$, $F_3$, and $F_4$ relative to the input $\mathbf{X}$ are 1, 1/2, 1/4, and 1/2, respectively. $SSR^m$ indicates $m$ consecutive SSR blocks. We adopt RS convolution as the feature down-sampling layer (Down) and sparse inverse convolution [40] as the feature up-sampling layer (UP). With an encoder-decoder structure, the SED block facilitates information exchange among spatially disconnected features, thereby enabling the model to capture long-range dependencies.

**DED block.** Existing high-performance 3D object detectors [15, 27, 28] usually rely on object centers for detection. However, the feature maps extracted by purely sparse CNNs may have empty holes around object centers, especially for large objects. To overcome this issue, we introduce a DED block that expands sparse features towards object centers, as shown in Figure 2 (b). The DED block shares a similar structure with the SED block but utilizes the widely used dense convolutions instead. Specifically, we replace the SSR block in the SED block with a dense residual (DR) block, which is similar to the SSR block but consists of two dense convolutions. Furthermore, the RS convolution employed for feature down-sampling is replaced with a DR block that has a stride of 2. For feature up-sampling, we replace the sparse inverse convolution with a dense deconvolution. These modifications enable the DED block to effectively expand sparse features towards object centers.

### 3.3 HEDNet

Based on the proposed SED block and DED block, we introduce HEDNet, a hierarchical encoder-decoder network designed for 3D object detection. The architecture of HEDNet is illustrated in Figure 3. Given the raw point clouds, a dynamic VFE module [41] is used to perform voxelization to generate a grid of voxels denoted as $F_0$. Subsequently, a sparse backbone including two SSR blocks and several SED blocks is employed to extract 3D sparse features. Before being fed into the 2D dense backbone, the sparse features are compressed into dense BEV features like in [18]. The 2D dense backbone, composed of $n$ DED blocks, is responsible for expanding the sparse features towards object centers. Finally, the output features are fed into the detection head for final predictions. At a macro level, HEDNet follows a hierarchical structure similar to SECOND [18], where the resolution of feature maps progressively decreases. At a micro level, the SED and DED blocks, key components of HEDNet, employ encoder-decoder structures. This is where the name HEDNet comes from. We adopt SED and DED blocks of three scales for HEDNet by default.

## 4 Experiments

### 4.1 Datasets and metrics

**Waymo Open** contains 160k, 40k, and 30k annotated samples for training, validation, and testing, respectively. The metrics for 3D object detection include mean average precision (mAP) and mAP

| Method | mAP/mAPH L2 | Vehicle AP/APH | | Pedestrian AP/APH | | Cyclist AP/APH | |
| --- | --- | --- | --- | --- | --- | --- | --- |
| | | L1 | L2 | L1 | L2 | L1 | L2 |

*Results on the validation data set*

| Method | mAP/mAPH L2 | Vehicle AP/APH L1 | L2 | Pedestrian AP/APH L1 | L2 | Cyclist AP/APH L1 | L2 |
| --- | --- | --- | --- | --- | --- | --- | --- |
| SECOND [18] | 61.0/57.2 | 72.3/71.7 | 63.9/63.3 | 68.7/58.2 | 60.7/51.3 | 60.6/59.3 | 58.3/57.0 |
| PointPillar [19] | 62.8/57.8 | 72.1/71.5 | 63.6/63.1 | 70.6/56.7 | 62.8/50.3 | 64.4/62.3 | 61.9/59.9 |
| Lidar-RCNN [42][†] | 65.8/61.3 | 76.0/75.5 | 68.3/67.9 | 71.2/58.7 | 63.1/51.7 | 68.6/66.9 | 66.1/64.4 |
| Part-A2-Net [37][†] | 66.9/63.8 | 77.1/76.5 | 68.5/68.0 | 75.2/66.9 | 66.2/58.6 | 68.6/67.4 | 66.1/64.9 |
| SST [14] | 67.8/64.6 | 74.2/73.8 | 65.5/65.1 | 78.7/69.6 | 70.0/61.7 | 70.7/69.6 | 68.0/66.9 |
| CenterPoint [27] | 68.2/65.8 | 74.2/73.6 | 66.2/65.7 | 76.6/70.5 | 68.8/63.2 | 72.3/71.1 | 69.7/68.5 |
| PV-RCNN [43][†] | 69.6/67.2 | 78.0/77.5 | 69.4/69.0 | 79.2/73.0 | 70.4/64.7 | 71.5/70.3 | 69.0/67.8 |
| CenterPoint [27][†] | 69.8/67.6 | 76.6/76.0 | 68.9/68.4 | 79.0/73.4 | 71.0/65.8 | 72.1/71.0 | 69.5/68.5 |
| SWFormer [13] | -/- | 77.8/77.3 | 69.2/68.8 | 80.9/72.7 | 72.5/64.9 | -/- | -/- |
| OcTr [44] | 70.7/68.2 | 78.1/77.6 | 69.8/69.3 | 80.8/74.4 | 72.5/66.5 | 72.6/71.5 | 69.9/68.9 |
| PillarNet-34 [11] | 71.0/68.5 | 79.1/78.6 | 70.9/70.5 | 80.6/74.0 | 72.3/66.2 | 72.3/71.2 | 69.7/68.7 |
| AFDetV2 [45] | 71.0/68.8 | 77.6/77.1 | 69.7/69.2 | 80.2/74.6 | 72.2/67.0 | 73.7/72.7 | 71.0/70.1 |
| CenterFormer [46] | 71.1/68.9 | 75.0/74.4 | 69.9/69.4 | 78.6/73.0 | 73.6/68.3 | 72.3/71.3 | 69.8/68.8 |
| LargeKernel3D[33] | -/- | 78.1/77.6 | 69.8/69.4 | -/- | -/- | -/- | -/- |
| PV-RCNN++ [47][†] | 71.7/69.5 | 79.3/78.8 | 70.6/70.2 | 81.3/76.3 | 73.2/68.0 | 73.7/72.7 | 71.2/70.2 |
| FSD [48][†] | 72.7/70.5 | 79.5/79.0 | 70.3/69.9 | 83.6/78.2 | 74.4/69.4 | 75.3/74.1 | 73.3/72.1 |
| DSVT-Voxel [15] | 74.0/72.1 | 79.7/79.3 | 71.4/71.0 | 83.7/78.9 | 76.1/71.5 | 77.5/76.5 | 74.6/73.7 |
| HEDNet (ours) | **75.3/73.4** | 81.1/80.6 | **73.2/72.7** | 84.4/80.0 | 76.8/72.6 | 78.7/77.7 | 75.8/74.9 |

*Results on the test data set*

| Method | mAP/mAPH L2 | Vehicle AP/APH L1 | L2 | Pedestrian AP/APH L1 | L2 | Cyclist AP/APH L1 | L2 |
| --- | --- | --- | --- | --- | --- | --- | --- |
| PV-RCNN [43] | 71.3/68.8 | 80.6/80.1 | 72.8/72.4 | 78.2/72.0 | 71.8/66.0 | 71.8/70.4 | 69.1/67.8 |
| PV-RCNN++ [47] | 72.4/70.2 | 81.6/81.2 | 73.9/73.5 | 80.4/75.0 | 74.1/69.0 | 71.9/70.8 | 69.3/68.2 |
| AFDetV2 [45] | 72.2/70.3 | 80.5/80.0 | 73.0/72.6 | 79.8/74.3 | 73.7/68.6 | 72.4/71.2 | 69.8/69.7 |
| FSD [48] | 74.4/72.4 | 82.7/82.3 | 74.4/74.1 | 82.9/77.9 | 75.9/71.3 | 75.6/74.4 | 72.9/71.8 |
| HEDNet (ours) | **76.9/75.0** | 84.2/83.8 | **77.0/76.6** | 84.1/79.7 | 78.3/74.0 | 78.2/77.0 | 75.4/74.3 |

Table 1: Comparison with prior methods on the Waymo Open dataset (single-frame setting). Metrics: mAP/mAPH (%)↑ for the overall results, and AP/APH (%)↑ for each category. [†]: two-stage method.

weighted by the heading accuracy (mAPH). Both are further broken down into two difficulty levels: L1 for objects with more than five LiDAR points and L2 for objects with at least one LiDAR point.

**nuScenes** consists of 28k, 6k, and 6k annotated samples for training, validation, and testing, respectively. Mean average precision (mAP) and nuScenes detection score (NDS) are used as the evaluation metrics. mAP is computed by averaging over the distance thresholds of 0.5m, 1m, 2m, 4m across all categories. NDS is a weighted average of mAP and the other five true positive metrics measuring the translation, scaling, orientation, velocity, and attribute errors.

### 4.2 Implementation details

We implemented our method using the open-source OpenPCDet [49]. To build HEDNet, we set the hyperparameter $m$ to 2 for all SED and DED blocks and stacked 4 DED blocks for the 2D dense backbone by default. For 3D object detection on the Waymo Open dataset, we adopted the detection head of CenterPoint and set the voxel size to (0.08m, 0.08m, 0.15m). We trained HEDNet for 24 epochs on the full training set (*single-frame*) to compare with prior methods. For ablation experiments in Section 4.4, we trained the models for 30 epochs on a 20% training subset. All models were trained with a batch size of 16 on 8 RTX 3090 GPUs. The other training settings strictly followed DSVT [15]. For 3D object detection on the nuScenes dataset, we adopted the detection head of TransFusion-L and set the voxel size to (0.075m, 0.075m, 0.2m). We trained HEDNet for 20 epochs with a batch size of 16 on 8 RTX 3090 GPUs. The other training settings strictly followed TransFusion-L [28].

### 4.3 Comparison with state-of-the-art methods

**Results on the Waymo Open dataset.** We compared the proposed HEDNet with previous methods on the Waymo Open dataset (Table 1). On the validation set, HEDNet yielded 1.3% L2 mAP and 1.3%

| Results on the validation data set | | | | | | | | | | | | |
|---|---|---|---|---|---|---|---|---|---|---|---|---|
| Method | NDS | mAP | Car | Truck | Bus | T.L. | C.V. | Ped. | M.T. | Bike | T.C. | B.R. |
| CenterPoint [27] | 66.5 | 59.2 | 84.9 | 57.4 | 70.7 | 38.1 | 16.9 | 85.1 | 59.0 | 42.0 | 69.8 | 68.3 |
| VoxelNeXt [50] | 66.7 | 60.5 | 83.9 | 55.5 | 70.5 | 38.1 | 21.1 | 84.6 | 62.8 | 50.0 | 69.4 | 69.4 |
| TransFusion-L [28] | 70.1 | 65.5 | 86.9 | 60.8 | 73.1 | 43.4 | 25.2 | 87.5 | 72.9 | 57.3 | 77.2 | 70.3 |
| HEDNet (Ours) | **71.4** | **66.7** | 87.7 | 60.6 | **77.8** | **50.7** | **28.9** | 87.1 | 74.3 | 56.8 | 76.3 | 66.9 |
| Results on the test data set | | | | | | | | | | | | |
| Method | NDS | mAP | Car | Truck | Bus | T.L. | C.V. | Ped. | M.T. | Bike | T.C. | B.R. |
| PointPillars [19] | 45.3 | 30.5 | 68.4 | 23.0 | 28.2 | 23.4 | 4.1 | 59.7 | 27.4 | 1.1 | 30.8 | 38.9 |
| 3DSSD [3] | 56.4 | 42.6 | 81.2 | 47.2 | 61.4 | 30.5 | 12.6 | 70.2 | 36.0 | 8.6 | 31.1 | 47.9 |
| CBGS [51] | 63.3 | 52.8 | 81.1 | 48.5 | 54.9 | 42.9 | 10.5 | 80.1 | 51.5 | 22.3 | 70.9 | 65.7 |
| CenterPoint [27] | 65.5 | 58.0 | 84.6 | 51.0 | 60.2 | 53.2 | 17.5 | 83.4 | 53.7 | 28.7 | 76.7 | 70.9 |
| FCOS-LiDAR [9] | 65.7 | 60.2 | 82.2 | 47.7 | 52.9 | 48.8 | 28.8 | 84.5 | 68.0 | 39.0 | 79.2 | 70.7 |
| HotSpotNet [52] | 66.0 | 59.3 | 83.1 | 50.9 | 56.4 | 53.3 | 23.0 | 81.3 | 63.5 | 36.6 | 73.0 | 71.6 |
| CVCNET [53] | 66.6 | 58.2 | 82.6 | 49.5 | 59.4 | 51.1 | 16.2 | 83.0 | 61.8 | 38.8 | 69.7 | 69.7 |
| AFDetV2 [45] | 68.5 | 62.4 | 86.3 | 54.2 | 62.5 | 58.9 | 26.7 | 85.8 | 63.8 | 34.3 | 80.1 | 71.0 |
| UVTR-L [54] | 69.7 | 63.9 | 86.3 | 52.2 | 62.8 | 59.7 | 33.7 | 84.5 | 68.8 | 41.1 | 74.7 | 74.9 |
| VISTA [55] | 69.8 | 63.0 | 84.4 | 55.1 | 63.7 | 54.2 | 25.1 | 82.8 | 70.0 | 45.4 | 78.5 | 71.4 |
| Focals Conv [56] | 70.0 | 63.8 | 86.7 | 56.3 | 67.7 | 59.5 | 23.8 | 87.5 | 64.5 | 36.3 | 81.4 | 74.1 |
| VoxelNeXt [50] | 70.0 | 64.5 | 84.6 | 53.0 | 64.7 | 55.8 | 28.7 | 85.8 | 73.2 | 45.7 | 79.0 | 74.6 |
| TransFusion-L [28] | 70.2 | 65.5 | 86.2 | 56.7 | 66.3 | 58.8 | 28.2 | 86.1 | 68.3 | 44.2 | 82.0 | 78.2 |
| LargeKernel3D [12] | 70.6 | 65.4 | 85.5 | 53.8 | 64.4 | 59.5 | 29.7 | 85.9 | 72.7 | 46.8 | 79.9 | 75.5 |
| LinK [16] | 71.0 | 66.3 | 86.1 | 55.7 | 65.7 | 62.1 | 30.9 | 85.8 | 73.5 | 47.5 | 80.4 | 75.5 |
| HEDNet (Ours) | **72.0** | **67.7** | 87.1 | 56.5 | 70.4 | **63.5** | **33.6** | 87.9 | 70.4 | 44.8 | 85.1 | 78.1 |

Table 2: Comparison with prior methods on the nuScenes dataset. Metrics: NDS (%)↑ and mAP (%)↑ for the overall results, AP (%)↑ for each category. 'T.L.', 'C.V.', 'Ped.', 'M.T.', 'T.C.', and 'B.R.' denote trailer, construction vehicle, pedestrian, motor, traffic cone, and barrier, respectively.

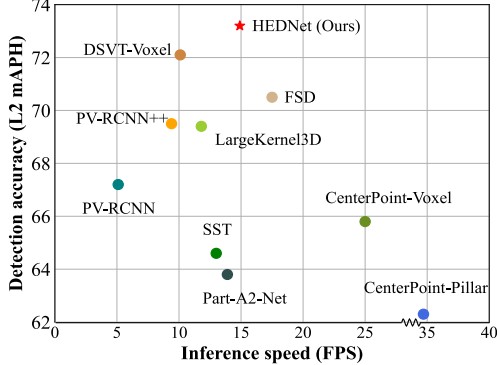

Figure 4: Detection accuracy (L2 mAPH) versus inference speed (FPS↑) of different models on the Waymo Open *validation* set.

Figure 5: Improvements decomposition of HED-Net over HEDNet-single with regard to the distance range of objects to the LiDAR sensor.

L2 mAPH improvements over the prior best method DSVT-Voxel [15]. HEDNet also outperformed the two-stage models PV-RCNN++ [47] and FSD [48]. More importantly, our method significantly outperformed the transformer-based DSVT-Voxel by 1.7% L2 mAPH on the vehicle category, where the average scale of vehicles is 10 times bigger than pedestrians and cyclists.

**Results on the nuScenes dataset.** We compared HEDNet with previous top-performing methods on the nuScenes dataset (Table 2). On the nuScenes test set, HEDNet achieved impressive results with 72.0% NDS and 67.7% mAP. Compared with TransFusion-L (which adopts the same head as HEDNet), HEDNet showcased significant improvements, with a gain of 1.8% NDS and 2.2% mAP. In addition, on the three categories with large objects, namely bus, trailer (T.L.), and construction vehicle (C.V.), HEDNet outperformed TransFusion-L by 4.1%, 4.7%, and 5.4% mAP, respectively. These results further demonstrate the effectiveness of our method.

| Block | Latency | L1 mAPH | L2 mAPH |
|---|---|---|---|
| RSR block | 176 ms | 74.61 | 68.30 |
| SSR block | 43 ms | 74.42 | 67.93 |
| SED block | 48 ms | 76.13 | 69.89 |
| SSR block[†] | 49 ms | 76.67 | 70.49 |
| SED block[†] | 54 ms | 77.39 | 71.37 |

(a) Effectiveness of the SED block.

| #Block | Latency | L1 mAPH | L2 mAPH |
|---|---|---|---|
| 0 | 48 ms | 76.13 | 69.89 |
| 1 | 54 ms | 77.56 | 71.37 |
| 3 | 63 ms | 77.75 | 71.64 |
| 4 | 67 ms | 78.02 | 71.92 |
| 5 | 73 ms | 77.85 | 71.77 |

(b) Effectiveness of the DED block.

| Sparse back. | Dense back. | L1 mAPH | L2 mAPH |
|---|---|---|---|
| 8 SSR blocks | 1 DED block | 73.91 | 67.80 |
| 16 SSR blocks | 1 DED block | 73.95 | 67.82 |
| 4 SED blocks | 1 DED block | 75.39 | 69.42 |
| 4 SED blocks | 2 DED blocks | 75.62 | 69.67 |

(c) HEDNet with 2D sparse backbone.

| #Scale | Latency | L1 mAPH | L2 mAPH |
|---|---|---|---|
| 1 | 43 ms | 76.18 | 69.88 |
| 2 | 59 ms | 77.61 | 71.44 |
| 3 | 67 ms | 78.02 | 71.92 |
| 4 | 78 ms | 78.12 | 72.02 |

(d) Effectiveness of encoder-decoder design.

Table 3: Ablations on the Waymo Open. [†]: with 1 DED block. In (c), 'back.' denotes backbone. In (d), the gray line denotes the HEDNet-single, and the blue line denotes the default HEDNet.

**Inference speed.** We further compared HEDNet with previous leading methods in terms of detection accuracy and inference speed, as depicted in Figure 4. Remarkably, HEDNet achieved superior detection accuracy compared with LargeKernel3D [12] and DSVT-Voxel [15] with faster inference speed. Note that LargeKernel3D and DSVT-Voxel were developed based on large-kernel CNN and transformer, respectively. All models were evaluated on the same NVIDIA RTX 3090 GPU.

## 4.4 Ablation studies

To better investigate the effectiveness of HEDNet, we constructed two network variants: HEDNet-single and HEDNet-2D. For the HEDNet-single, we replaced all the SED and DED blocks in HEDNet with single-scale blocks, *i.e.,*only keeping the first $m$ SSR/DR blocks in each SED/DED block. For the HEDNet-2D, we replaced all 3D sparse convolutions in HEDNet with 2D sparse convolutions and removed the three feature down-sampling layers that follow the feature maps $F_1$, $F_2$, and $F_3$, following the settings in DSVT [15]. The two SSR blocks after $F_0$ were also removed. In HEDNet-2D, the resolution of the output feature map of the 2D dense backbone is same as that of the network input $F_0$. We conducted experiments on the Waymo Open dataset to analyze various design choices of HEDNet. All models were trained on a 20% training subset and evaluated on the validation set.

### 4.4.1 Model designs

**Effectiveness of the SED block.** We compared the models built with RSR block, SSR block, and our proposed SED block in Table 3 (a). For the models with RSR/SSR blocks, we replaced the SED blocks in HEDNet with RSR/SSR blocks. The 2D dense backbones in the first three models were removed to fully explore the potential of the three structures. The first model with RSR blocks achieved slightly better results than the second model with SSR blocks but with much higher runtime latency. The third model with SED blocks significantly outperformed the second model with SSR blocks by 1.96% L2 mAPH. Similar gains can be observed in the last two models with DED blocks.

**Effectiveness of the DED block.** The DED block is designed to expand sparse features towards object centers. We compared the models that include different numbers of DED blocks in Table 3(b). The models with DED blocks achieved large improvements over the model without DED blocks. The model with five blocks performed worse than the model with four blocks. The former may be overfitted to the training data. We adopted four DED blocks for HEDNet by default.

**HEDNet with 2D sparse backbone.** We conducted experiments on HEDNet-2D to evaluate the effectiveness of our method with 2D inputs. For the construction of 2D inputs, we set the voxel size to (0.32m, 0.32m, 6m), where the size of 6m in the Z axis corresponds to the full size of the input point clouds. To compare our SED blocks with SSR blocks, we replaced each SED block in HEDNet-2D with 2 SSR blocks or 4 SSR blocks, resulting in two models of different sizes (the first two models in

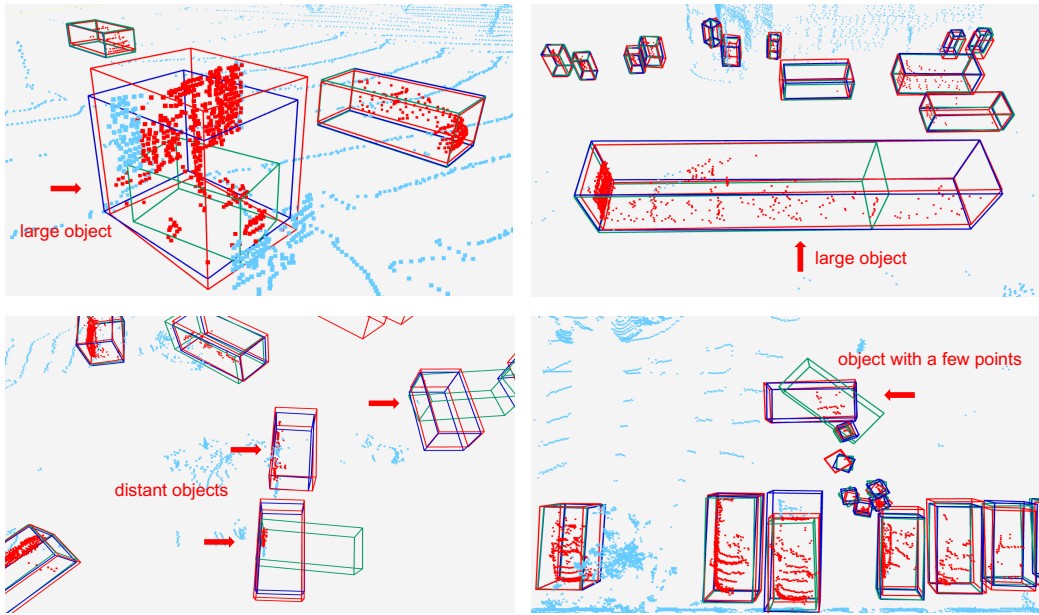

Figure 6: Qualitative results on the Waymo Open. The red boxes are annotated by humans. The blue boxes and green boxes are predicted by HEDNet and the HEDNet-single, respectively. Red points correspond to the points that fall inside the human-annotated boxes. HEDNet predicted more precise bounding boxes for the objects marked by red arrows than the single-scale variant HEDNet-single.

Table 3 (c)). From Table 3 (c), we can make the following observations. Firstly, the model with 16 SSR blocks achieved similar performance to the model with 8 SSR blocks, indicating that *stacking more SSR blocks could not further boost performance*. Secondly, the models incorporating SED blocks showed significant improvements over the models using SSR blocks (at least 1.6% gains on L2 mAPH). This observation demonstrates the effectiveness of our SED block. Thirdly, stacking two DED blocks achieved better performance than using a single one. These results clearly demonstrate the generality and effectiveness of our proposed SED block and DED block.

### 4.4.2 HEDNet versus HEDNet-single

We conducted a thorough comparison between the proposed HEDNet and its single-scale variant, HEDNet-single, to explore the effectiveness of the encoder-decoder structure and investigate which objects benefit from HEDNet the most. Please note that the HEDNet is designed to capture long-range dependencies among features in the spatial space, which is the core of this work.

**Firstly,** we compared the models built with blocks of different numbers of scales to explore the effectiveness of the encoder-decoder structure. As shown in Table 3 (d), the models with multi-scale blocks significantly outperformed the single-scale variant HEDNet-single (the line in gray color). Using more scales achieved better performance, but introduced higher runtime latency. To strike a balance between accuracy and efficiency, we adopted three-scale blocks for HEDNet by default.

**Secondly,** we evaluated the three-scale HEDNet and the HEDNet-single in Table 3 (d) separately for each category and analyzed the results based on the distance range of objects to the LiDAR sensor. We illustrate the accuracy improvements of HEDNet over HEDNet-single at various distance ranges in Figure 5. Firstly, HEDNet showed significant improvements over HEDNet-single on the vehicle category, where the size of vehicles is 10 times larger than that of pedestrians and cyclists. This highlights the importance of capturing long-range dependencies for accurately detecting large objects. Furthermore, HEDNet achieved larger performance gains on distant objects compared with objects closer to the LiDAR sensor across all three categories. We believe this is because distant objects with fewer point clouds require more contextual information for accurate detection. Overall, these results demonstrate the effectiveness of our proposed method in detecting large and distant objects.

**Thirdly,** we further present some visualization results of the two models in Figure 6. HEDNet-single exhibited limitations in accurately predicting boxes for large objects and the predicted boxes often

only covered parts of the objects (see the top row). In addition, when dealing with objects containing a few points, HEDNet-single struggled to accurately estimate their orientations (see the bottom row). In contrast, HEDNet predicted more precise bounding boxes for both scenarios, which we believe is owed to the ability of HEDNet to capture long-range dependencies.

## 5    Conclusion

We propose a sparse encoder-decoder structure named SED block to capture long-range dependencies among features in the spatial space. Further, we propose a dense encoder-decoder structure named DED block to expand sparse features towards object centers. With the SED and DED blocks, we introduce a hierarchical encoder-decoder network named HEDNet for 3D object detection in point clouds. HEDNet achieved a new state-of-the-art performance on both the Waymo Open and nuScenes datasets, which demonstrates the effectiveness of our method. We hope that our work can provide some inspiration for the backbone design in 3D object detection.

**Limitations**    HEDNet mainly focuses on 3D object detection in outdoor autonomous driving scenarios. However, the application of HEDNet in other indoor applications is still an open problem.

**Acknowledgements**    This work was supported in part by the National Key Research and Development Program of China (No. 2021ZD0200301) and the National Natural Science Foundation of China (Nos. U19B2034, 61836014) and THU-Bosch JCML center.

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

| Method | Car | Bicycle | Motorcycle | Truck | Bus | Person | Bicyclist | Motorcyclist | Road | Parking | Sidewalk | Other ground | Building | Fence | Vegetation | Trunk | Terrian | Pole | Traffic sign | mIoU |
|---|---|---|---|---|---|---|---|---|---|---|---|---|---|---|---|---|---|---|---|---|
| MinkUNet34 [58] | 96.8 | 55.0 | 81.4 | 83.2 | 70.2 | 79.5 | 89.8 | 7.8 | 94.8 | 54.6 | 82.8 | 1.5 | 92.0 | 68.3 | 87.9 | 69.4 | 72.8 | 66.1 | 52.4 | 68.3 |
| HEDNet (Ours) | 97.3 | 57.2 | 82.3 | 88.1 | 73.9 | 80.4 | 91.3 | 23.2 | 95.1 | 51.5 | 83.1 | 2.8 | 92.1 | 69.6 | 87.6 | 69.4 | 72.3 | 66.6 | 52.1 | **70.3** |

Table 4: 3D semantic segmentation results on the SemanticKiTTI validation set. Metrics: mIoU (%)↑ for the overall results, IoU (%)↑ for each category.

## A    Implementation details on 3D object detection

We implemented our method with Pytorch using the open-source OpenPCDet [49].

**Waymo Open dataset.**   We set the hyperparameter $m$ to 2 for all SED and DED blocks and stacked 4 DED blocks for the 2D dense backbone by default. We adopted the detection head of CenterPoint [27] for HEDNet. As metioned in the main paper, we primarily followed the training and inference schemes of DSVT [15]. Specifically, the voxel size was set to (0.08m, 0.08m, 0.15m), and the detection range was set to [-75.2m, 75.2m] for X and Y axis, and [-2m, 4m] for Z axis. We trained HEDNet for 24 epochs on the entire training dataset and reported the evaluation results on the validation set to compare with previous state-of-the-art methods. For the ablation experiments, we trained all models for 30 epochs on a 20% training subset. All models were trained with a batch size of 16 on 8 RTX 3090 GPUs. We employed the Adam [57] optimizer with a one-cycle learning rate policy, and set the weight-decay to 0.05, and the max learning rate to 0.003. We also adopted the faded training strategy in the last epoch. During inference, we applied class-specific NMS with an IoU threshold of 0.75, 0.6 and 0.55 for vehicle, pedestrian, and cyclist, respectively.

**nuScenes dataset.**   We set the hyperparameter $m$ to 2 for all SED and DED blocks and stacked 5 DED blocks for the 2D dense backbone. We adopted the detection head of TransFusion-L [28] for HEDNet and primarily followed the training and inference schemes of TransFusion-L [28]. The voxel size was set to (0.075m, 0.075m, 0.2m), and the detection range was set to [-54m, 54m] for X and Y axis, and [-5m, 3m] for Z axis. We trained HEDNet for 20 epochs on the combined training and validation sets with a batch size of 16 on 8 RTX 3090 GPUs and reported the results on the test set to compare with other methods. We employed the Adam [57] optimizer with a one-cycle learning rate policy, and set the weight-decay to 0.1, the momentum to [0.85, 0.95], and the max learning rate to 0.001. The faded strategy was used during the last 5 epochs. For submission to the test server, we set the query number of detection head to 300 and did not use any test-time augmentation.

## B    Experiments on 3D semantic segmentation

We conducted experiments on the popular LiDAR semantic segmentation dataset SemanticKiTTI [59], It provides 22 sequences with 19 semantic classes, captured by a 64-beam LiDAR sensor. Following the standard practice [58, 60], we report the Intersection-over-Union (IoU) for each category and the average score (mIoU) over all categories. For the backbone network, we employed a UNet-style structure, *i.e.,*the same designs as the first two layers of the MinkUNet34 are first adopted to extract sparse features with a spatial down-sampling ratio of 4, followed by 4 SED layers to transform the resulting features, finally, two symmetrical layers are used to recover high-resolution features following MinkUNet34. The other settings strictly followed [58]. Table 4 shows that the proposed model exhibited significant gains over its counterpart MinkUNet34 (*i.e.,*2.0% in mIoU), which demonstrated the generality of our method.

## C    A step-wise ablation from VoxelNet to HEDNet

We conducted a step-wise ablation from the standard VoxelNet to our HEDNet on the Waymo Open dataset to show the effectiveness of different components (see Table 5). For the second model, we employed the training tricks used by DSVT, including IoU loss, class-specific NMS, faded strategy (disabling data augmentations in the last epoch), and a weight decay of 0.05. These training tricks

| No. | VoxelNet | Tricks* | Smaller-voxel | SED-block | DED-block | Full-data | Latency | L1 mAPH | L2 mAPH |
|---|---|---|---|---|---|---|---|---|---|
| 1 | ✓ | | | | | | 40 ms | 70.0 | 64.0 |
| 2 | ✓ | ✓ | | | | | 40 ms | 75.4 | 69.1 |
| 3 | ✓ | ✓ | ✓ | | | | 49 ms | 76.6 | 70.2 |
| 4 | ✓ | ✓ | ✓ | ✓ | | | 55 ms | 77.6 | 71.3 |
| 5 | ✓ | ✓ | ✓ | ✓ | ✓ | | 67 ms | 78.0 | 71.9 |
| 6 | ✓ | ✓ | ✓ | ✓ | ✓ | ✓ | 67 ms | 79.5 | 73.4 |

Table 5: A step-wise ablation from VoxelNet to HEDNet. The first five models were trained on a 20% training subset and the last model was trained on the full training set. *: training tricks used in DSVT.

can significantly boost detection accuracy. Actually, most of the training tricks have been used by previous works, such as PV-RCNN++, and FSD. The codes and training configurations of the DSVT model can be found in the OpenPCDet archives. For the third model, we adopt a smaller input voxel size to keep more detailed information, which boosts the detection accuracy of pedestrian and cyclist. The 4th and 5th models sequentially incorporate our proposed SED blocks and DED blocks. The last model was trained on the full training set. Our final model (the 6th model) outperformed the previous SOTA method DSVT by 1.3% L2 mAPH while being 50% faster.

