# OpenReview forum: "HEDNet: A Hierarchical Encoder-Decoder Network for 3D Object Detection in Point Clouds"
_NeurIPS.cc/2023/Conference — NeurIPS 2023 poster_

### Official Review · Reviewer_pAfm · 2023-06-24

**Soundness:** 3 good
**Presentation:** 3 good
**Contribution:** 2 fair
**Rating:** 6
**Confidence:** 5

**Summary:**

This paper focuses on 3D object detection using outdoor LiDAR data. The authors proposed a multi-scale network that is built upon several encoder-decoder blocks, e.g., SED and DED. The motivation behind it is to use sub-manifold sparse feature down-sampling and up-sampling steps to capture long-range voxel relationships without greatly sacrificing computation efficiency. The authors benchmark their model, HEDNet, on both Waymo Open Dataset and nuScenes and demonstrate competitive detection quality.

**Strengths:**

[Strong Benchmark Performance]

1. This paper demonstrates strong detection quality on standard benchmarks. For example, HEDNet outperforms the prior SOTA DSVT while being 50% faster.

[Method Novelty]

2. Essentially, the network architecture plotted in Figure 3 can be thought as some type of “hierarchical stack-hourglass” models. Each block of SED and DED modules is a tiny "hourglass" with skip connections.  In that sense, HEDNet differentiates itself from other 3D object detectors which mostly follow FPN structures (e.g., SWFormer) or single-stride designs (e.g., SST). I agree that HEDNet brings some fresh air into this field.

**Weaknesses:**

[Lack of Insights or Theoretical Contributions]

1. While acknowledging the method novelties of HEDNet as mentioned above, I also think this work might lack insights or theoretical contributions. For example, I’m not sure the findings/designs are something that is sufficient for a NeurIPS publication or is generalizable to other 3D deep learning tasks. This paper is a good fit for other conferences like CVPR or ICCV.

**Questions:**

[Technical Clarity]

1. L-146 to L-147: “we adopt RS convolution as the feature down-sampling layer (DOWN)”. However, Figure 2 (a) says that the down-sampling layer is SS conv, instead of RS conv as explained in the text. They are conflicting with each other. Which one is correct?

2. Table-1: are the presented results all based on single frame or multiple frames? This is not mentioned in the main paper.

**Limitations:**

N/A.

---

> ### Author Rebuttal · Authors · 2023-08-08
>
> #### Q1. Might lack of insights or theoretical contributions? Not sure the findings/designs are generalizable to other 3D deep learning tasks?
>
> To the best of our knowledge, not all works published at NeurIPS prioritize theoretical contributions. In fact, NeurIPS has been a platform for the publication of numerous works that offer valuable insights. For instance, within the domain of image object detection and segmentation, notable publications include Faster R-CNN (NeurIPS 2016), RFCN (NeurIPS 2016), SOLOv2 (NeurIPS 2020), and MaskFormer (NeurIPS 2021). When it comes to 3D object detection, recently published methods such as Object DGCNN (NeurIPS 2021), FSD (NeurIPS 2022), DeepInteraction (NeurIPS 2022), and BEVFusion (NeurIPS 2022) stand out.
>
> The essence of our paper lies in the introduction of the encoder-decoder block, which offers a simple yet effective approach to capturing long-range dependencies for LiDAR-based 3D vision tasks. We sincerely appreciate the insightful feedback from you and the other two reviewers, k2sW and vKEU. As a result, we've integrated the proposed encoder-decoder design into the LiDAR-based 3D semantic segmentation task, yielding a model that achieved excellent performance. For more detailed information, please refer to Q4 of the reviewer vKEU.
>
> #### Q2. Figure 2 (a) says that the down-sampling layer is SS conv?
> The down-sampling layer is RS conv and the description in L146-147 is correct. We made a mistake in Figure 2(a) and we apologize for the caused confusion. We will update the paper to fix the error.
>
> #### Q3. Table-1: are the presented results all based on a single frame or multiple frames?
> Single frame. We will update the paper to clarify this.

---

> > ### Comment · Reviewer_pAfm · 2023-08-14
> >
> > Many thanks to the authors for addressing my questions! I will maintain my rating and have no objections to accepting this paper.

---

> > > ### Author Response · Authors · 2023-08-15
> > > **Rely to Reviewer pAfm**
> > >
> > > Thank you very much for your dedicated efforts to review our paper and suggestions for our work.

---

### Official Review · Reviewer_Nfby · 2023-07-04

**Soundness:** 2 fair
**Presentation:** 3 good
**Contribution:** 4 excellent
**Rating:** 7
**Confidence:** 5

**Summary:**

The paper proposed a Hierarchical Encoder-Decoder Network, which faciliated sparse encoder-decoder block to enhance the network to model long-range dependencies among sparse voxel features while maintaining high sparsity. The method consistently improve the detection performance with a higher detection rate.

**Strengths:**

1. The idea is simple and efficient.
2. Clearly identify the issue among current detection model w.r.t the submanifold sparse conv and regular sparse conv.


**Weaknesses:**

1. Since the authors adopt a 3×3 RS convolution with a stride of 3 for feature down-sampling (Down), this will definitely introduce more non-zero elements in the resulted downsampled feature. In relation to the original convolutional block in SECOND, which involves performing SSxN + RS(down), I am curious about how the SED block, encompassing SS + RS(down) + SS + RS(down) + SS + UP + UP, can effectively reduce the computation and maintain the sparsity.



**Questions:**

It is nice to witness a simple idea being effectively work within3D object detection. My major concern revolve around the technical soundness of this encoder-decoder design and its ability to truly preserve sparsity and save computation.

It would be better to anonymously publish the results in the NuScenes and Waymo leaderboard.

**Limitations:**

The author addressed the limitations.

---

> ### Author Rebuttal · Authors · 2023-08-08
>
> #### Q1. How the SED block can effectively reduce the computation and maintain the sparsity?
> 1. As you said, each SED block encompasses SS + RS_1(down) + SS + RS_2(down) + SS + UP_2 + UP_1. To distinguish different Down and Up layers, we add subscripts here. We adopt the sparse inverse conv implemented in the open source toolboxes [spconv](https://github.com/traveller59/spconv) as the UP layer, and it is the key for SED block to maintain feature sparsity. Each UP layer is bound with the Down layer which has the same subscript, and it only upsamples features to the same non-zero positions as that of the features before the bound Down layer. As a result, the input and output features of each SED block have the same feature sparsity.
>
> 2. The sparse backbone of SECOND includes several convolutional blocks, where each block involves performing SSxN + RS(down). Our proposed SED block is designed to replace the N consecutive SS layers before the RS layer. Compared with the sparse backbone in SECOND, our design has a slightly higher computation cost (only 5ms, see Table 3 (a) in the paper, row 2 *vs.* row 3). But our final HEDNet is still more efficient than previous methods that incorporate large kernel convolutions or self-attention mechanisms, such as LargeKernel3D and DSVT, while achieving higher detection accuracy. (see Figure 4 in the paper)
>
> #### Q2. It would be better to anonymously publish the results in the NuScenes and Waymo leaderboards.
> We have anonymously published the results in the NuScenes and Waymo leaderboards and sent the links to AC following the rules of the rebuttal. The results on the nuScenes leaderboard were the same as that presented in Table 2 of the paper. The results on the Waymo leaderboard were inferred from the test set by the model presented in Table 1 of the paper (only used the training set for training, based on single-frame data). As we are not sure whether you can see the link, we present the results on the Waymo leaderboard as follows:
>
> ||||||||
> |-|-|-|-|-|-|-|
> |L2 mAPH|Veh/L1 APH|Veh/L2 APH|Ped/L1 APH|Ped/L2 APH|Cyc/L1 APH|Cyc/L2 APH|
> |73.8|83.4|76.0|79.0|73.3|74.8|72.1|
>
> Note: L2 mAPH is the metric for the overall results.

---

> > ### Comment · Reviewer_Nfby · 2023-08-16
> >
> > Thanks for your response. The authors address my major concern and again it is nice to see simple idea works well. I would raise my score to 7 (accept).

---

> > > ### Author Response · Authors · 2023-08-16
> > > **Rely to Reviewer Nfby**
> > >
> > > We sincerely appreciate the time and effort you dedicated to reviewing our paper, and we are deeply honored to have our work recognized by you.

---

### Official Review · Reviewer_vKEU · 2023-07-05

**Soundness:** 4 excellent
**Presentation:** 4 excellent
**Contribution:** 3 good
**Rating:** 7
**Confidence:** 5

**Summary:**

This paper focuses on the long range problems in 3D object detection. Compared to the common idea of large kernel or transformers, this hierarchical design is more smart and efficient. It designs an encoder-decoder style network for 3D object detection. Comprehensive experiments on nuScenes and Ways show the effectiveness of the proposed methods. Detailed ablation studies show the efficiency compared to large kernel and transformer baselines.

**Strengths:**

1. This paper is well-written. The idea is direct and easy to follow. Figures and tables are well-organized.

2. The idea of encoder-decoder is also much smart. Compared to large kernel or transformers, this design seems more relevant to the task 3D object detection itself. That is also why it is more efficient than baselines. This simple design might be motivated from semantic segmentation in 2D images.

3. Experimental results on nuScenes and Waymo are state-of-the-art. These are good to show the effectiveness of the proposed methods.

**Weaknesses:**

1. The visualization comparison on receptive fields should be provided. As this work claimed that it achieves similar or better long-range detection capability than large kernel and transformers. This visualization is necessary to support this claim.

2. A kind suggestion is that the results on nuScenes validation set should also be released. This would facilitate the following works to compare with.

3. Fully sparse styles are gradually becoming popular, as FSD and VoxelNeXt. It would be curious that whether the idea will work if there are only sparse SED. Because the dense head limits the detection range, (like Argoverse2 datasets with 200m radius), as analyzed in FSD. It would be much helpful that if this work could be extended to be fully sparse.

4. This idea is very related to the area of semantic segmentation. I think this framework might simultaneously work with lidar semantic segmentation,  with different head. For example, like LIdarMultiNet, training and inference together on both detection and segmentation on nuScenes and waymo. I know that this suggestion might not be validated well in the limited rebuttal period. Please just ignore this if you do not have enough time for this.

**Questions:**

1. If the authors choose to skip the experiments on the suggestion of 3/4 in the weakness. Please at least provide a discussion on them, which would be helpful to further improve the quality of this paper.

**Limitations:**

It is a bit not elegant that the author implement this idea upon two separate codebases on different datasets (OpenPCDet to Waymo, mmdet3d to nuScenes). A kind suggestion is that you could unified the codebase when you released the code. For example, you could reproduce the results of nuScenes on OpenPCDet, and use the unified one to release.

---

> ### Author Rebuttal · Authors · 2023-08-08
>
> #### Q1. The visualization comparison on receptive fields should be provided. As this work claimed that it achieves similar or better long-range detection capability than large kernel and transformers.
>
> We have presented some visualization examples in the provided PDF file (refer to `Author Rebuttal by Authors`, the global rebuttal filed above). To illustrate receptive fields, we back-propagated gradients from the output BEV feature that corresponds to each object center. In each sub-figure, the red points indicate the receptive field corresponding to the selected object center.
>
> We conducted a comparison among the proposed HEDNet, HEDNet-single, and DSVT to underline the effectiveness of our approach. HEDNet-single is a single-scale variant of HEDNet, employing a backbone akin to the standard sparse backbone (SECOND). Further details about HEDNet-single can be found in the first paragraph of Section 4.4 of the paper. DSVT, the prior state-of-the-art method for 3D object detection on the Waymo Open dataset, utilizes a transformer-based backbone. As LargeKernel3D exhibits inferior performance compared with DSVT, we only consider DSVT here.
>
> The visualized examples highlight that both HEDNet and DSVT can capture longer-range dependencies than HEDNet-single. Additionally, HEDNet achieves similar or slightly larger receptive fields than DSVT. To better bolster our claim, we calculated the average distance from the farthest K red points to their corresponding object center. With K set to 30, the average distance ranking across the three models was as follows: HEDNet (38.8m) > DSVT (28.2m) > HEDNet-single (22.1m). These results demonstrate that our proposed method exhibits better long-range detection capabilities than HEDNet-single and DSVT. All analyses were conducted on the Waymo validation set.
>
> #### Q2. Results on nuScenes validation set
>
> |||||||||||||||
> |-|-|-|-|-|-|-|-|-|-|-|-|-|-|
> |No.|Model|NDS|mAP|Car|Truck|Bus|Trailer|Cons.|Ped.|Motor.|Bicycle|Tr.Cone|Barrier|
> |1|TransFusion-L|70.2|65.2|86.5|60.8|25.7|74.2|41.6|70.2|71.6|56.8|87.1|77.1|
> |2|HEDNet (Ours)|71.4|66.9|87.8|61.4|27.7|77.6|48.0|69.9|72.6|58.9|87.2|77.6|
>
> The first model is our baseline TransFusion-L, which employs the standard 3D sparse backbone. The second model is the proposed HEDNet. The two models were trained with the same settings, and they only differ in their backbones.
>
> #### Q3. Experimental results on VoxelNeXt.
>
> ||||||||||
> |-|-|-|-|-|-|-|-|-|
> |No.|Model|L2 mAPH|Veh/L1 APH|Veh/L2 APH|Ped/L1 APH|Ped/L2 APH|Cyc/L1 APH|Cyc/L2 APH|
> |1|VoxelNeXt|70.8|77.6|69.3|77.9|70.4|75.4|72.6|
> |2|Our model|78.2|72.1|79.8|71.6|78.3|70.8|76.7|73.9|
> ||+improvements|+1.3|**+2.2**|**+2.3**|+0.4|+0.4|+1.3|+1.3|
>
> Note: L2 mAPH is the metric for the overall results.
>
> We agree with your opinion about the fully sparse detectors and conducted experiments with VoxelNeXt on the 100% Waymo Open dataset. We replaced the 3D sparse backbone of VoxelNeXt with our proposed sparse backbone. Our model (the second model) outperformed the standard VoxelNeXt by 1.3% L2 mAPH, demonstrating that our method also performed well on fully sparse detectors.
>
> #### Q4. Experimental results on LiDAR-based 3D semantic segmentation.
>
> ||||||||||||||
> |-|-|-|-|-|-|-|-|-|-|-|-|-|
> |No.|Model|mIoU|Motorcyclist|Truck|Bicycle|Motorcycle|Bicyclist|Trunk|Person|Sidewalk|Traffic sign|Building|
> |1|MinkUNet34|68.3|9.9|80.2|52.8|80.4|89.8|68.3|79.3|83.1|52.3|92.3|
> |2|Our model|70.3|23.2|88.1|57.2|82.3|91.3|69.4|80.4|83.1|52.1|92.1|
> ||+improvements|+2.0|+13.3|+7.9|+4.4|+1.9|+1.5|+1.1|+1.1|+0.0|-0.2|-0.2|
>
> Note: mIoU is the metric for the overall results.
>
> Due to the limited rebuttal period, we only conducted LiDAR-based 3D semantic segmentation experiments on the SemanticKITTI dataset, which has a smaller size than the nuScenes and Waymo Open datasets. We adopted the well-known model MinkUNet implemented in MMDetection3d as our baseline (the first model). For the second model, we replaced the sparse UNet of the first model with three SED blocks. Both models were trained with the same settings and only differ in their backbones. Our model outperformed the plain MinkUNet by 2.0% mIoU. This validates the effectiveness of our design on LiDAR-based 3D semantic segmentation.
>
> #### Q5. Unify the codebase?
> Thanks for your suggestion. We will unify the codebase to release our codes.

---

> > ### Comment · Reviewer_vKEU · 2023-08-12
> > **Reply to Authors**
> >
> > Many thanks for your detailed reply. My concerns have been well-addressed by these experiments. This work is solid enough to be accepted. I will raise my rate to "7 Accept". Please remember to release code and models, this is important and beneficial for this area.

---

> > > ### Author Response · Authors · 2023-08-14
> > > **Rely to Reviewer vKEU**
> > >
> > > Thanks a lot for your insightful feedback and recognition of our work. We will release the code and models.

---

### Official Review · Reviewer_zRCr · 2023-07-06

**Soundness:** 4 excellent
**Presentation:** 4 excellent
**Contribution:** 4 excellent
**Rating:** 7
**Confidence:** 5

**Summary:**

The proposed method considers a much more straightforward approach to incorporating larger context for 3D detection from point clouds. In contrast to existing work which consider large kernels or transformers to increase the receptive field of 3D backbones, this work focuses on developing an encoder-decoder block, taking care to maintain input and output sparsity levels. The resulting architecture achieves new state-of-the-art on LiDAR-only Waymo and nuScenes benchmarks with high efficiency.

**Strengths:**

- The paper is well-written, and the relevant modules are illustrated clearly.
- The resulting pipeline achieves strong performance while being efficient and straightforward, not involving complicated modules.
- Extensive ablations validate the importance of the SED block, as well as the # of DED blocks.


**Weaknesses:**

- While there is some analysis of what types of objects improved with the scale in the encoder-decoder block, it would be useful to see a more fine-grained analysis on nuScenes, which contains more object categories.
- Table 3a’s row 2 appears to be similar, or same, as the standard 3D sparse convolutional backbone used in prior works. In that case, the overall model should be similar to the CenterPoint model, which on 20% data on OpenPCDet archives (70.76 + 65.49 + 67.39) / 3 = 67.88~ L1 mAPH. Why is the performance in Table 3a row 2 much better, at 74.42? Are there other differences between the two models?
- Building on the previous point, it would be useful to see a step-by-step ablation from the standard 3D sparse convolutional backbone to the proposed backbone, to demonstrate more clearly where the improvements/latency changes are with respect to the most commonly used 3D spconv backbone.

**Questions:**

Please see Weaknesses above. Overall, the proposed method has solid performance with high efficiency, opting for a more straightforward backbone that demonstrates better performance than existing large-kernel or transformer structures. As such, I recommend acceptance at this stage.

**Limitations:**

Limitations have been included.

---

> ### Author Rebuttal · Authors · 2023-08-08
>
> #### Q1. A more fine-grained analysis on nuScenes?
>
> |||||||||||||
> |-|-|-|-|-|-|-|-|-|-|-|-|
> |Model|mAP|Car|Truck|Cons.veh|Bus|Trailer|Barrier|Motor.|Bicycle|Ped.|Tr.Cone|
> |TransFusion-L|65.2|86.5|60.8|25.7|74.2|41.6|70.2|71.6|56.8|87.1|77.1|
> |HEDNet (Ours)|66.9|87.8|61.4|27.7|77.6|48.0|69.9|72.6|58.9|87.2|77.6|
> |||||||||||||
> |+improvements|+1.7|+1.3|+0.6|**+2.0**|**+3.4**|**+6.4**|-0.3|+1.0|+2.0|+0.1|+0.5|
> | avg object size ($m^3$)||16.0|56.0|73.0|115.4|151.2|1.3|2.3|1.4|0.9|0.2|
>
> Note: The column mAP shows the overall results, and the other columns show AP of each category.
>
> Since our baseline model on the nuScenes dataset (TransFusion-L) uses a sparse backbone similar to that of HEDNet-single, we directly compare HEDNet with TransFusion-L. The two models only differ in their backbones.
> 1) We present the detailed results on the nuScenes validation set in the above table. HEDNet significantly outperformed TransFusion-L on large objects, especially the categories Trailer, Bus, and Construction vehicle.
> 2) In addition, we divided objects into three groups by object distances, *i.e.* 0-15, 15-30, and 30+ meters. HEDNet outperformed TransFusion-L by 1.6%, 2.0%, and 2.8% mAP in the three groups, respectively, indicating that HEDNet benefits the detection of distant objects more.
>
> #### Q2. A step-by-step ablation from the standard 3D sparse convolutional backbone to the proposed backbone?
>
> |||||||||||
> |-|-|-|-|-|-|-|-|-|-|
> |No.|Model|FPS|L2 mAPH|Veh/L1 APH|Veh/L2 APH|Ped/L1 APH|Ped/L2 APH|Cyc/L1 APH|Cyc/L2 APH|
> |1|CenterPoint (ResNet)|25.0 |64.0|72.2|64.4|68.0|60.3|69.8|67.3|
> |2|+DSVT tricks|25.0|69.1|76.5|68.2|75.8|67.9|73.8|71.1|
> |3|+Smaller voxel|20.4|70.2|77.1|68.7|77.3|69.6|75.2|72.4|
> |4|+SED blocks|18.1|71.3|78.4|70.2|78.3|70.8|75.9|73.1|
> |5|+DED blocks|15.0|71.9|79.5|71.4|78.2|70.8|76.4|73.6|
> |6|+100% data|15.0|73.4|80.6|72.7|80.0|72.6|77.7|74.9|
> ||||||||||||
> |*|DSVT-Voxel (100%)|10.1|72.1|79.3|71.0|78.9|71.5|76.5|73.7|
>
> Note: L2 mAPH is the metric for the overall results. FPS denotes frame per second, the metric for measuring inference speed.
>
> We conducted a step-by-step ablation on the Waymo Open dataset (20% training data for the first five models, and 100% training data for the last two models).
> 1. The results of the first model are from the OpenPCDet archives.
> 2. The second model employed the training tricks used by DSVT, including IoU-aware loss, multi-class NMS, faded strategy (disabling data augmentations in the last epoch), and a bigger weight decay value. These training tricks can significantly boost detection accuracy. Actually, most of the training tricks have been used by previous works, such as PV-RCNN++, and FSD. The codes and training configurations of the DSVT model can also be found in the OpenPCDet archives.
> 3. For the third model, we adopt a smaller input voxel size to keep more detailed information, which boosts the detection accuracy of pedestrian and cyclist.
> 4. The 4th and 5th models sequentially incorporate our proposed SED blocks and DED blocks.
> 5. The last two models were trained with the full training set. Our final model (the 6th model) outperformed the previous SOTA method DSVT by 1.3% L2 mAPH while being 50% faster.
>
> #### Q3. The performance of the model in Table 3a row 2 is much better than the CenterPoint model on OpenPCDet archives?
> All models in Table 3 were trained with the aforementioned settings (DSVT tricks and smaller input voxel size), thus significantly outperforming the models released on the OpenPCDet archives.

---

> > ### Comment · Reviewer_zRCr · 2023-08-20
> >
> > I thank the authors for their careful responses. It is good to see that the proposed method improves performance on larger objects, as expected, without hurting smaller objects. I hope that the step-by-step ablation can also be included in the supplementary, as it would be valuable for decoupling improvement from tricks vs improvement from modules in prior work as well.
> >
> > As such, I maintain my original rating.

---

> > > ### Author Response · Authors · 2023-08-20
> > > **Rely to Reviewer zRCr**
> > >
> > > Thank you very much for your recognition of our work. We will include the step-by-step ablation in the supplementary materials to help readers better understand our work.

---

### Official Review · Reviewer_k2sW · 2023-07-10

**Soundness:** 3 good
**Presentation:** 3 good
**Contribution:** 3 good
**Rating:** 6
**Confidence:** 2

**Summary:**

The paper addresses the task of 3D object detention on real-world 3D inputs that possess challenges such as scalability to long-range dependency. The main suggestion is to incorporate encoder-decoder type blocks to capture long-range dependency, in contrast to previous works that were limited to the sparsity patterns in the input. The method is evaluated on 2 real-world 3D detection benchmarks.

**Strengths:**

The paper is well-written and easy to follow. The introduction discusses previous approaches, their limitations, and how the current suggestion can mitigate them.

The method is evaluated on real-world challenging benchmarks.

I appreciate the incorporation of the qualitative evaluation and the ablation study.

The proposed solution seems to be simple and valuable.


**Weaknesses:**

Missing discussion

It is not clear what part of the design choices stemmed from the fact the architecture is aimed to tackle object detection. It would be beneficial to incorporate such as discussion. In this context, the comment on the limitation section is a bit vague.


**Questions:**

No specific quesitons

**Limitations:**

Yes

---

> ### Author Rebuttal · Authors · 2023-08-08
>
> #### Q1. It is not clear what part of the design choices stemmed from the fact the architecture is aimed to tackle object detection? And the comment on the limitation section is a bit vague?
> Since we only conducted experiments on 3D object detection in the paper, we didn't know whether the proposed HEDNet was suitable for other 3D tasks. This is the limitation we want to express. As reminded by you and the other two reviewers vKEU and pAFm, we realize that HEDNet can actually work on other 3D tasks. During the rebuttal period, we conducted experiments on the LiDAR-based 3D semantic segmentation task. The results show that HEDNet also performed well on semantic segmentation, effectively addressing the aforementioned limitation. For more detailed information, please refer to Q4 of the reviewer vKEU.

---

> > ### Comment · Reviewer_k2sW · 2023-08-21
> > **rebuttal**
> >
> > Thank you for the rebuttal. I have no further concerns.

---

### Author Rebuttal · Authors · 2023-08-08

To All Reviewers:
Thank you so much for your affirmations and insightful feedback on our work. We are very glad to receive your constructive suggestions. Here, we give more discussions and explanations on those concerns mentioned by each reviewer respectively.

To Reviewer vKEU:
The visualization comparison on receptive fields is presented in the attached PDF file.

---

### Author Response · Authors · 2023-08-09
**Anonymized links required by Reviewer Nfby (Q2)**

1. [Link](https://eval.ai/web/challenges/challenge-page/356/leaderboard/1012) to the results on the nuScenes leaderboard.

https://eval.ai/web/challenges/challenge-page/356/leaderboard/1012

2. [Link](https://waymo.com/open/challenges/entry/?challenge=DETECTION_3D&challengeId=DETECTION_3D&emailId=ae022fcb-9223&timestamp=1691590461963005) to the results on the Waymo leaderboard.

https://waymo.com/open/challenges/entry/?challenge=DETECTION_3D&challengeId=DETECTION_3D&emailId=ae022fcb-9223&timestamp=1691590461963005

---

### Decision · Program_Chairs · 2023-09-21

**Decision:**

Accept (poster)

**Comment:**

The paper is motivated to address the limitation of submanifold sparse conv. and proposes a hierarchical encoder-decoder network for 3D object detection, which is able to capture long-range dependencies among features in the spatial space. Experiments on the Waymo and nuScenes datasets show the efficacy of the proposed method.

Reviewers are initially concerned about some paper presentations and visualizations, about the lack of insights such as analysis on how and where the proposed method improves 3D detection,  and about the applicability of the method on scenarios other than self-driving cars. The authors provide responses that address these concerns. Congratulations on acceptance of the paper!